# Automated detection scheme for acute myocardial infarction using convolutional neural network and long short-term memory

**Ryosuke Muraki**[1], **Atsushi Teramoto**[2]*, **Keiko Sugimoto**[3], **Kunihiko Sugimoto**[4], **Akira Yamada**[5], **Eiichi Watanabe**[5]

**1** Graduate School of Health Sciences, Fujita Health University, Toyoake, Japan, **2** Faculty of Radiological Technology, School of Medical Sciences, Fujita Health University, Toyoake, Japan, **3** Faculty of Medical Technology, School of Medical Sciences, Toyoake, Japan, **4** Fujita Health University Hospital, Toyoake, Japan, **5** School of Medicine, Fujita Health University, Toyoake, Japan

* teramoto@fujita-hu.ac.jp

**Data Availability Statement:** Some of image data and numerical data are deposited in Figshare at the

## Abstract

The early detection of acute myocardial infarction, which is caused by lifestyle-related risk factors, is essential because it can lead to chronic heart failure or sudden death. Echocardiography, among the most common methods used to detect acute myocardial infarction, is a noninvasive modality for the early diagnosis and assessment of abnormal wall motion. However, depending on disease range and severity, abnormal wall motion may be difficult to distinguish from normal myocardium. As abnormal wall motion can lead to fatal complications, high accuracy is required in its detection over time on echocardiography. This study aimed to develop an automatic detection method for acute myocardial infarction using convolutional neural networks (CNNs) and long short-term memory (LSTM) in echocardiography. The short-axis view (papillary muscle level) of one cardiac cycle and left ventricular long-axis view were input into VGG16, a CNN model, for feature extraction. Thereafter, LSTM was used to classify the cases as normal myocardium or acute myocardial infarction. The overall classification accuracy reached 85.1% for the left ventricular long-axis view and 83.2% for the short-axis view (papillary muscle level). These results suggest the usefulness of the proposed method for the detection of myocardial infarction using echocardiography.

## Introduction

Acute myocardial infarction (AMI) is a disease in which myocardial cells become necrotic due to thrombus formation or blood vessel occlusion. AMI causes severe chest pain and requires immediate treatment, such as percutaneous transluminal coronary recanalization or coronary artery bypass grafting. It is important to diagnose AMI as early as possible because it can lead to heart failure, arrhythmia, or sudden death.

Echocardiography, a noninvasive imaging modality used to diagnose AMI, enables the real-time assessment of cardiac function and complications and evaluation of regional abnormal wall motion in patients with AMI. Therefore, it is widely used in cardiology. However, depending on disease range and severity, regional abnormal wall motion can be difficult to recognize. Moreover, the accuracy of its recognition depends on sonographer experience.

following URL: https://doi.org/10.6084/m9.figshare.16530876.

**Funding:** This research was funded by Grants-in-Aid for Scientific Research from Japan Society for the Promotion of Science (https://www.jsps.go.jp/english/e-grants/index.html, https://kaken.nii.ac.jp/en/grant/KAKENHI-PROJECT-21K08140/, Grant Number 21K08140, awarded to EW and AT) and Japan Agency for Medical Research and Development (https://www.amed.go.jp/en/index.html, Grant Number 20hk0102071h0001, awarded to EW). The funders had no role in study design, data collection and analysis, decision to publish, or preparation of the manuscript.

**Competing interests:** The authors have declared that no competing interests exist.

Deep learning, an artificial intelligence technique, was recently confirmed to have excellent processing power. Convolutional neural networks (CNNs), which are deep learning models, have been widely applied in various fields such as medical image analysis [1–12]. In the domain of echocardiography, Kusunose et al. proposed a method for detecting regional abnormal wall motion on echocardiography images and obtained a high detection accuracy with an area under the curve of approximately 0.9 [7]. Huang et al. proposed a technique for the visualization of AMI on echocardiography, and their results showed a Dice index of approximately 0.8 [9]. Thus, CNN is a highly accurate technique for AMI detection and classification.

Deep learning has also been widely applied for the multi-classification of video datasets. Recurrent neural networks (RNNs), which are also deep learning models, are particularly effective at predicting and classifying sequential data such as wave signals, natural language, and video [13–16]. RNNs have a recursive structure, meaning that they analyze and produce outputs based on previous time-series and sequential data. Long short-term memory (LSTM), which was recently developed, can be used to analyze even larger and longer-term datasets than RNNs can process [17, 18]. Methods based on LSTM have been widely applied, and Ullah et al. proposed a multi-classification method for video images in large video datasets that obtained a classification accuracy of > 90% [19]. Zhou et al. proposed a natural language classification method based on LSTM and obtained a classification accuracy of >80% [20]. In addition, there has been considerable research done on combining CNNs and RNNs [21].

Kusunose et al. conducted a study on a novel detection method using deep learning techniques for abnormal regional wall motion noted on echocardiography. Multiple deep learning models were utilized to classify echocardiography into routine normal cases and those with abnormal regional wall motion. They adopted the ROC curve as an evaluation method and compared the classification accuracy of the clinical technologists to that of the proposed deep learning model. This comparison resulted in the AUC of the proposed method (0.97) and the AUC of the clinical technologists (0.95) being virtually equivalent, confirming the effectiveness of the proposed method. However, the study had a few limitations. The first limitation was the use of short-axis echocardiogram views only at the level of the papillary muscle. Second, the deep learning input data were limited to only three images per case: end-diastolic, mid-systolic, and end-systolic. Therefore, we attempted to detect AMI by analyzing the wall motion temporal changes and inputting the echocardiograph of one cardiac cycle into deep learning, which can then analyze time series. In addition to the short-axis papillary muscle (PM) level view, we used another view that is easy to use, the left ventricular long axis view, to detect AMI. We believe this method could be integral to the development of detection support technology so that non-specialist physicians in their clinics can accurately detect an AMI. In this study, we focused on CNN and LSTM, which have been widely applied to medical image analysis and can be used for video processing and analysis. Specifically, we aimed to develop an automated detection scheme for AMI using CNN and LSTM in echocardiography.

## Materials and methods

### Overview of the proposed method

An overview of the proposed method is presented in Fig 1. Echocardiography images were loaded into VGG16 [22], a CNN model, to extract the features. The obtained features were then analyzed using LSTM for the classification of AMI and normal myocardium.

### Echocardiography

For this study, we collected short-axis PM level and left ventricular LX view images taken with ultrasound equipment (Vivid E9 and Vivid E95, GE Healthcare) at Fujita health university

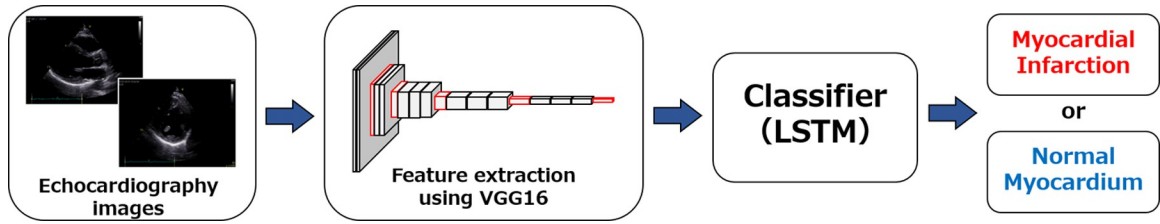

**Fig 1. Overview of the proposed method.**

hospital. A total of 202 cines were collected as inputs: 99 diagnoses of acute anteroseptal infarction of the proximal left anterior descending artery in the American Heart Association Committee Report and 103 normal cases. Cardiologists and experienced sonographers usually estimate the culprit coronary artery in patients with myocardial infarction using left ventricular long-axis, left ventricular short-axis, apical four-chamber, apical two-chamber, and apical long-axis views. Here we employed the short-axis PM level and left ventricular long-axis views because it is relatively easy to detect anteroseptal infarction of the proximal left anterior descending artery, which can be used to evaluate abnormal wall motion in cases of anteroseptal infarction. Moreover, all cases of anteroseptal infarction used in this study underwent coronary angiography, and occlusion of #6 was observed by examination. In addition, patients who underwent percutaneous coronary intervention after coronary angiography and had abnormal wall motion on echocardiography were included. Fig 2 shows the views used. Table 1 and Fig 3 show the baseline clinical characteristics of this cohort and the selection of the study

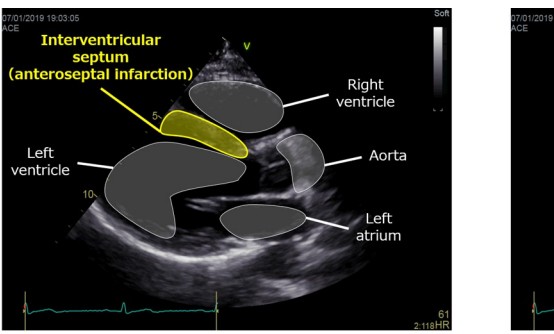
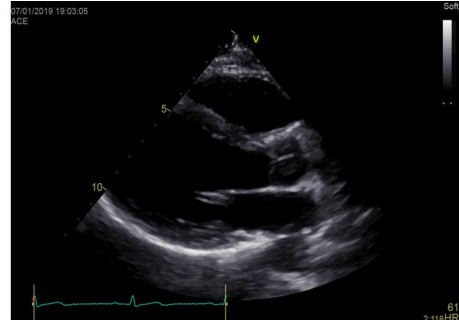

**Left ventricular long-axis view**

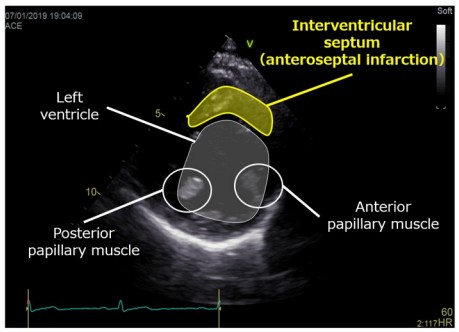
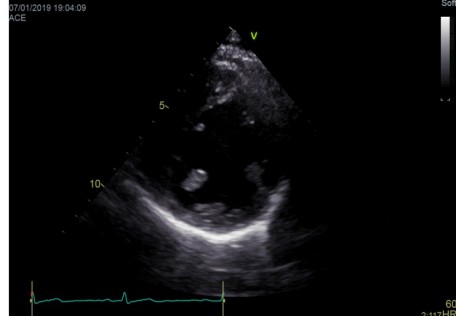

**Short-axis papillary muscle level view**

**Fig 2.** Left ventricular long-axis view and short-axis papillary muscle level view (left: views of anatomy; right: normal views).

**Table 1. Baseline clinical characteristics of the study cohort.**

| Characteristic | Control (n = 103) | AMI (n = 99) | P value |
|---|---|---|---|
| Age, mean (SD) | 66.7(12.1) | 69.4 (11.6) | 0.21 |
| Female sex, no. (%) | 38(38.4) | 24 (24.2) | 0.051 |
| Body mass index (kg/m$^2$), mean (SD) | 22.4(4.2) | 21.3(7.9) | 0.07 |
| Systolic blood pressure (mmHg), mean (SD) | 127.3(21.5) | 149.9 (29.6) | <0.001 |
| Diastolic blood pressure (mmHg), mean (SD) | 75.23(13.9) | 88.6 (20.8) | <0.001 |
| Cardiac rhythm | 76.7(13.5) | 84.1 (19.9) | 0.026 |
| Sinus rhythm, mean (SD) | | 83.1 (21.6) | |
| Atrial fibrillation, mean (SD) | | 85.1 (20.6) | |
| Site of AMI | | | |
| Anterior/lateral/inferior (%) | | Anterior (100) | |
| Coronary risk factors | | | |
| Hypertension, no. (%) | 4(4.04) | 58 (58.6) | <0.001 |
| Diabetes, no. (%) | | 30 (30.3) | |
| Dyslipidemia, no. (%) | | 47 (47.5) | |
| Smoking, no. (%) | | 57 (57.6) | |
| Laboratory data | | | |
| White blood cell count, mean (SD) | 6.7(3.5) | 10.1 (7.0) | <0.001 |
| Hemoglobin (g/dL), mean (SD) | 10.8(2.5) | 13.7 (12.9) | <0.001 |
| Creatinine (mg/dL), mean (SD) | 1.2(1.7) | 1.2(1.3) | <0.001 |
| eGFR (mL/min/1.73 m$^2$), mean (SD) | 78.4(32.1) | 82.3 (36.6) | 0.41 |
| BNP (pg/mL), mean (SD) | 85.8(144) | 391.8 (738.6) | 0.009 |
| CK-MB (ng/ml), mean (SD) | 5.8(5.9) | 280.9 (452.1) | <0.001 |
| TnI (ng/ml), mean (SD) | 0.2(0.6) | 89.3 (226.1) | <0.001 |
| Medications | | | |
| β-blocker, no. (%) | 16(15.1) | 63 (63.6) | <0.001 |
| ARB /ACE-I, no. (%) | 8(7.8) | 65 (65.6) | <0.001 |

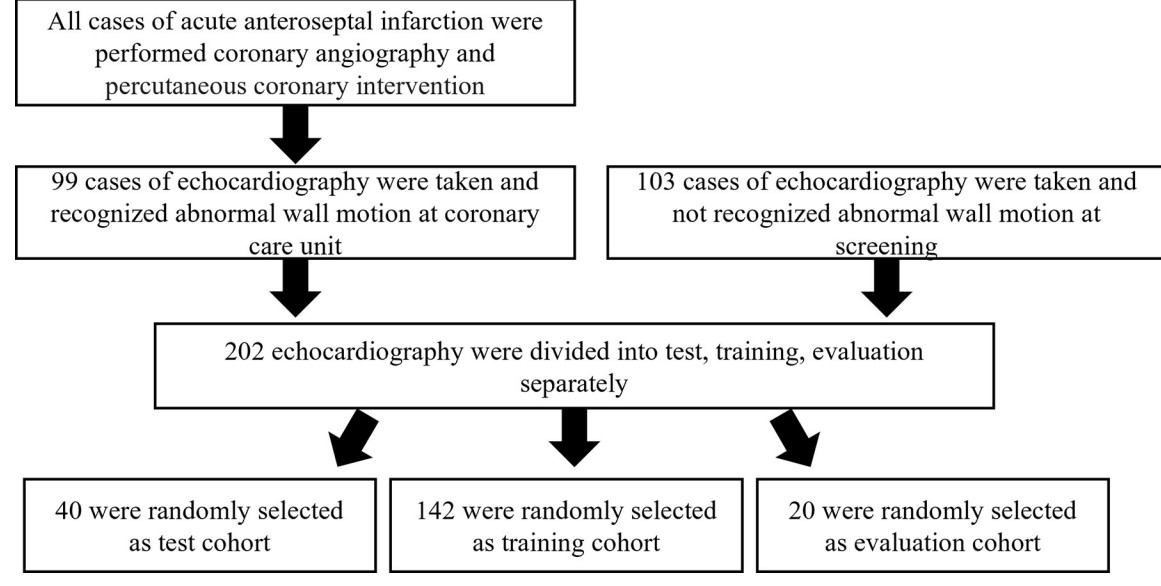

All cases of acute anteroseptal infarction were performed coronary angiography and percutaneous coronary intervention

99 cases of echocardiography were taken and recognized abnormal wall motion at coronary care unit

103 cases of echocardiography were taken and not recognized abnormal wall motion at screening

202 echocardiography were divided into test, training, evaluation separately

40 were randomly selected as test cohort

142 were randomly selected as training cohort

20 were randomly selected as evaluation cohort

**Fig 3. Selection of the study population.**

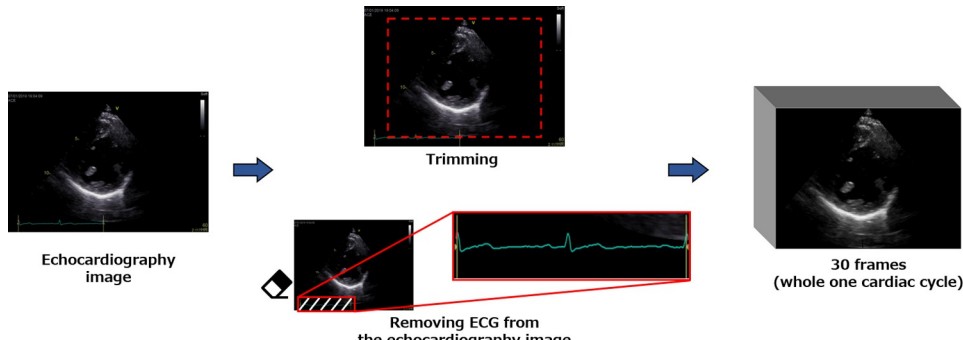

**Fig 4. Preprocessing of the input data.** All frames of one cardiac cycle taken from the original echocardiography image were interpolated to 30 frames, and the ECG in the image was removed and trimmed.

population, respectively. P values indicate differences between patients who have normal myocardium and AMI. $P < 0.05$ was considered statistically significant.

The image preprocessing involved electrocardiogram (ECG) removal, cropping, and frame interpolation of the views. On echocardiography, a two-lead ECG is drawn to identify indicators during the cardiac cycle such as end-diastole and end-systole. Fig 4 shows the preprocessing of the input images. In this study, to recognize the cardiac wall motion on each view, the ECG was removed and trimmed to form a bounding rectangle. To consider differences in heart rate between patients, one cardiac cycle was extracted from each image and the number of frames was interpolated to 30. Interpolation means that if the number of frames in the video image for one cardiac cycle was 50–60, they were interpolated at equal intervals so that the number of frames was 30, whereas if the number of frames was 10–20, they were interpolated so that the total number of echocardiography images for all patients was 30. Linear interpolation was used as the complementation method, and each of the images for one cardiac cycle was extracted based on the length between the peaks (R-R interval) of the simultaneously recorded ECG. This study was approved by an institutional review board of Fujita Health University and informed consents were obtained from patients subject to the condition of data anonymization (No. HM19-345).

## Feature extraction

The features of the interpolated images were extracted for input into the classification model [23]. CNNs can extract features simultaneously as the final outputs or extract parameters as intermediate outputs from individual layers. Varshni et al. used CNNs to extract features from chest radiographs [24]. Hyeon et al. used CNNs to extract features from cytology images and used conventional machine learning methods to differentiate between benign and malignant cells [24, 25]. By using CNNs as feature extractors and other models as output layers, new inputs can be added and the accuracy further improved compared to CNN use alone. Therefore, we focused on this method and adopted VGG16 and global pooling [26], another CNN model, as feature extractors. Using these feature extraction methods, we extracted features from all frames of the interpolated echocardiography images.

**VGG16 model.** This study used VGG16, a CNN model developed by the Visual Geometry Group at the University of Oxford in 2014, for the feature extraction. The structure of VGG16 is shown in Fig 5. It consists of 13 convolutional layers, 5 pooling layers, and 3 fully connected layers [27]. We introduced the VGG16 pretraining network using the large natural ImageNet image dataset. From the second fully connected layer of VGG16, 4096 features were extracted and input into the classification model.

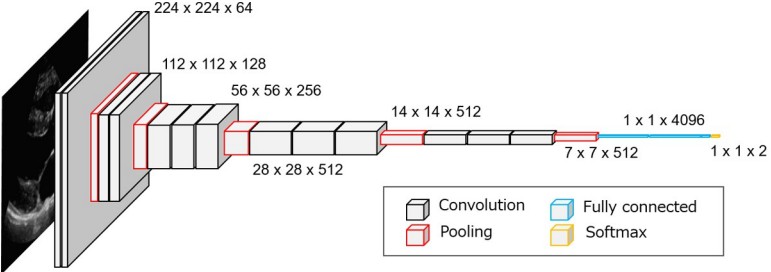

**Fig 5. VGG16 structure.**

**Global pooling.** Global average pooling (GAP) and global max pooling (GMP) are 2 compression methods for feature maps extracted from images by CNNs. These compression methods select only the maximum or average values from the last feature map of the CNN and pool all of the feature maps. Fig 6 shows a simplified diagram of the GAP and GMP methods. These compression methods were applied to the feature maps after completion of the convolutional and pooling processes until just before the fully connected layer of the CNN. GAP, shown in the upper part of Fig 6, extracts the average value from each feature map and outputs only the extracted value as an intermediate output. GMP, shown in the lower part of Fig 6, extracts the maximum value from each feature map and outputs it as an intermediate output. This process significantly reduces the number of dimensions from the original feature map parameters and prevents overfitting. In this study, we performed global pooling of the $7 \times 7 \times 512$ feature maps extracted by VGG16 and output the average or maximum value from each feature map, which resulted in a total of 512 parameters.

## LSTM networks

Because the detection of AMI on echocardiography requires the evaluation and analysis of wall motion over time, two-dimensional images with different time phases were input into the CNN. We also focused on RNNs, which are excellent tools for processing sequential data and effective for time-series information such as cine images and wave signals as well as text data and natural language. Therefore, this model is characterized by its ability to control sequential information. Fig 7 demonstrates the principle of RNN, where $x$, $y$, and $h$ are the input, output, and weight of the hidden layer, respectively.

The RNN connects the layer at time ($t$) with the previous layer ($t-1$) and calculates the parameters in the hidden layer ($h(t)$) and the output according to the following equations:

$$h(t) = f(Ux(t) + Wh(t-1)) \tag{1}$$

$$y(t) = g(Vh(t)) \tag{2}$$

$U$, $W$, and $V$ denote the weights calculated during training, and $f(z)$ and $g(z_m)$ denote the sigmoid and softmax functions, respectively. The equations for the respective activation functions are as follows:

$$f(z) = \frac{1}{1 + e^{-z}} \tag{3}$$

$$g(z_m) = \frac{e^{z_m}}{\sum_k + e^{z_k}} \tag{4}$$

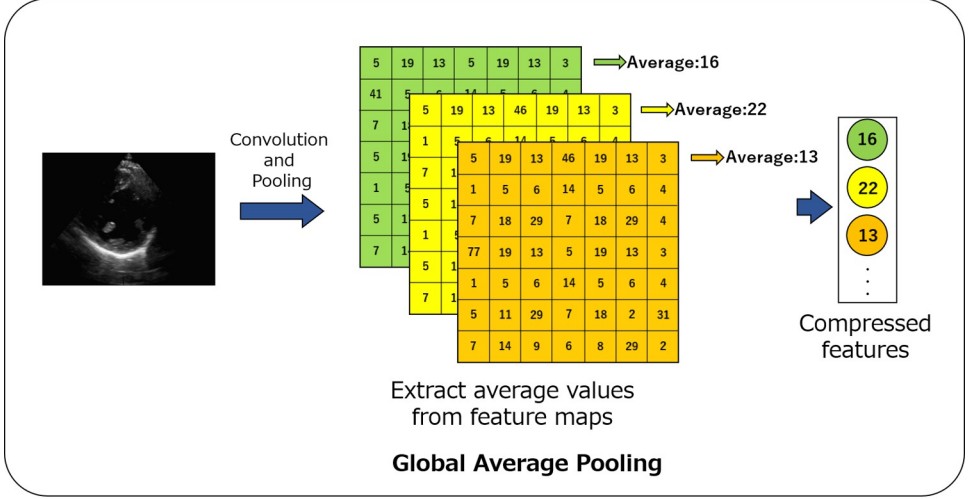

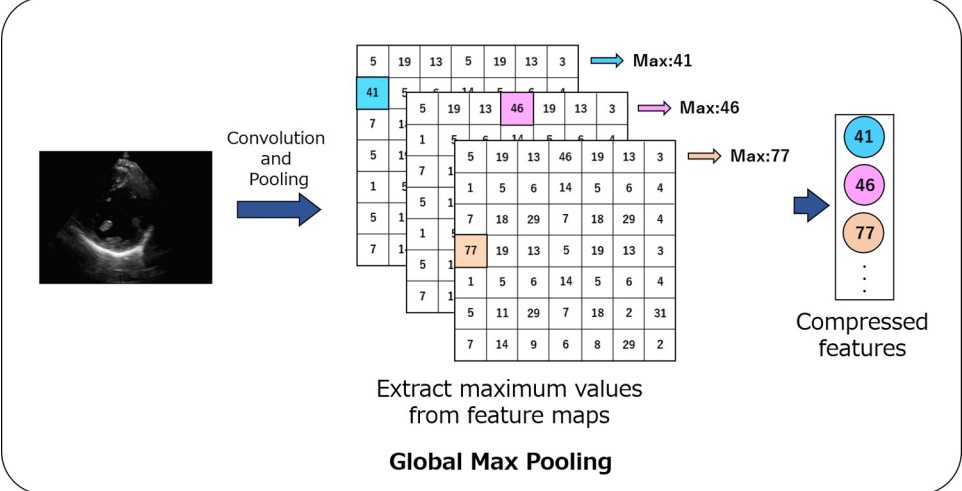

**Fig 6. Simplified diagram of the global average pooling and global max pooling compression methods.** Pooling is performed to extract the maximum or average values from the last feature map of the CNN.

However, because RNNs theoretically store all past data during training, the vanishing gradient problem arises due to the divergence and disappearance of weights. Therefore, we focused on LSTM, an improved RNN model. The principle of the LSTM is shown in Fig 8. The difference between LSTM and RNN is that LSTM features a mechanism of information selection called "gate" and "cell." There are 3 types of gates: "input," "output," and "forgetting." Eq (i) shows the formula for the forgetting gate ($f_t$):

$$f_t = \sigma(W_{xf}x_t + W_{hf}h_{t-1} + W_{Cf}C_{t-1} + b_f) \tag{i}$$

From the input information, the output of the LSTM layer at ($t$−1) and the cell, the information that is unnecessary in the learning at ($t$), is selected and "forgotten." Eq (ii) is used to determine the input gate ($I_t$):

$$I_t = \sigma(W_{xI}x_t + W_{hI}h_{t-1} + W_{CI}C_{t-1} + b_I) \tag{ii}$$

The output of the last LSTM layer and the value of the cell are used to determine the new value

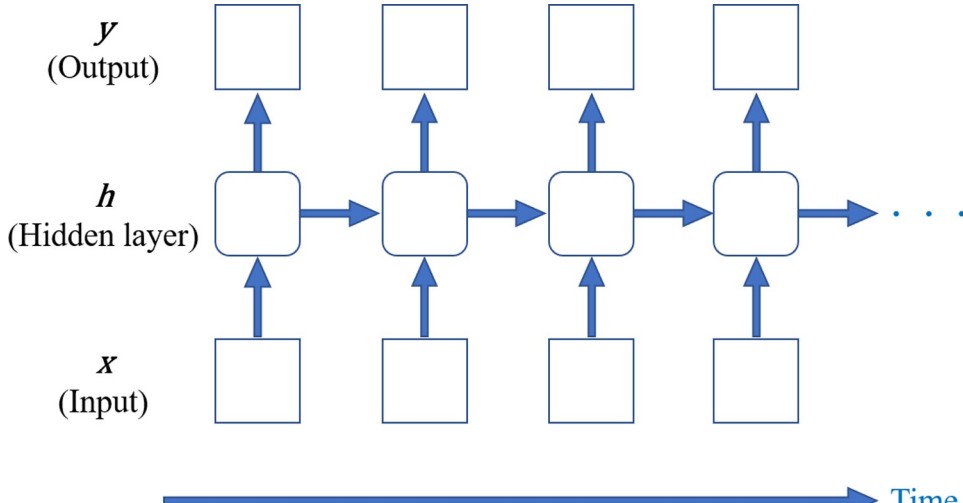

**Fig 7. Schematic diagram of a recurrent neural network.** In the diagram, *x* is the input, *h* is the hidden layer, and *y* is the output. The RNN learns by passing the weights of the hidden layer to the next hidden layer in the forward direction.

to be updated. The value of the updated cell ($C_t$) is then determined by Eq (iii):

$$C_t = f_t C_{t-1} + I_t \tanh(W_{xc} x_t + W_{hC} h_{t-1} + b_C) \tag{iii}$$

The value of the cell determined by the above equation is propagated to the next LSTM layer, and the output ($O_t$) of the LSTM layer at (*t*) is determined by Eqs (iv) and (v) in the output

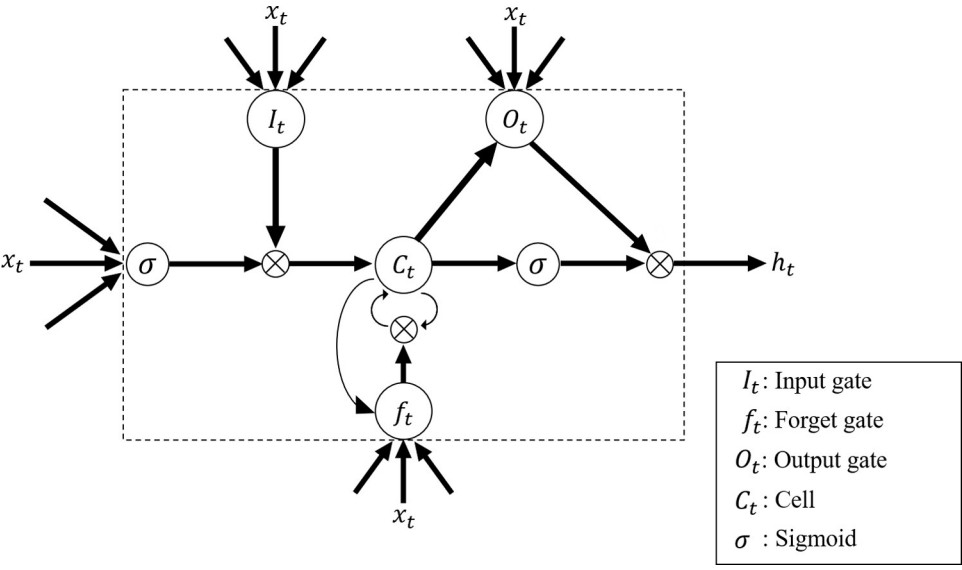

$I_t$ : Input gate
$f_t$ : Forget gate
$O_t$ : Output gate
$C_t$ : Cell
$\sigma$ : Sigmoid

**Fig 8. Diagram of the long short-term memory principle.** Input, forgetting, and output gates are used to determine the information to be input into, retained, and output from the cells, enabling the learning of sequential data over a long period of time.

gate section:

$$O_t = \sigma(W_{xO}x_t + W_{hO}h_{t-1} + W_{Cf}C_{t-1} + b_O) \qquad\qquad \text{(iv)}$$

$$h_t = O_t \tanh(C_t) \qquad\qquad \text{(v)}$$

Using this gating mechanism, LSTM can analyze long-term series data and solve the vanishing gradient problem of conventional RNNs. We introduced these mechanisms to analyze the wall motion over time using echocardiography.

Finally, as shown in Fig 9, the features extracted by the VGG16 method were input into the LSTM to classify the normal and AMI cases. For the hyperparameters, we set the learning rate to $1 \times 10^{-5}$, the number of epochs to 50, the batch size to 30, and the input data size to $4096 \times 30$.

## Evaluation

**Cross-validation method.** The cross-validation method was used to evaluate the classification accuracy of the constructed model. Fig 10 shows a simplified diagram of the cross-validation method. All datasets were initially divided into several groups, one of which was used as the test group for the evaluation, while the remaining data were used for training. Thereafter, the accuracy was comprehensively calculated by repeating the process such that all data are test data. In this study, we used a five-fold cross-validation method in which the 202 echocardiography images were divided into 142 cases for training, 20 cases for evaluation during training, and the remaining 40 cases for testing, with random sampling so that all cases were used as test data.

**Comparison with conventional artificial neural network.** To demonstrate the effectiveness of our method, the classification accuracy was also evaluated using five-fold cross-validation on a normal artificial neural network (ANN), which does not feature a mechanism to handle time-series relations separately from LSTM [28, 29]. A schematic of the ANN is shown in Fig 11, in which all features extracted by the VGG16 method for each frame were combined

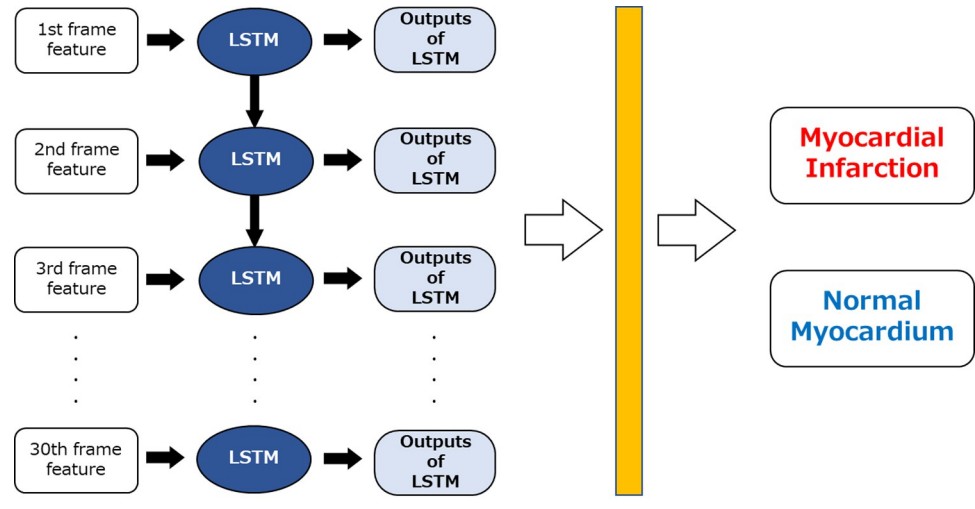

**Fig 9. Classification of myocardial infarction and normal myocardium cases. LSTM, long short-term memory.**
Features from 1 to 30 frames were input to each LSTM, and the classification was performed using the softmax function based on the LSTM output.

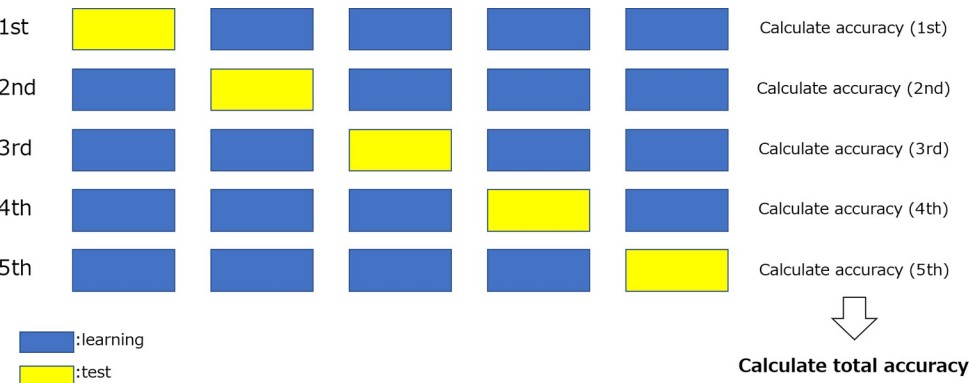

**Fig 10. Simplified diagram of the cross-validation method (number of folds = 5).**

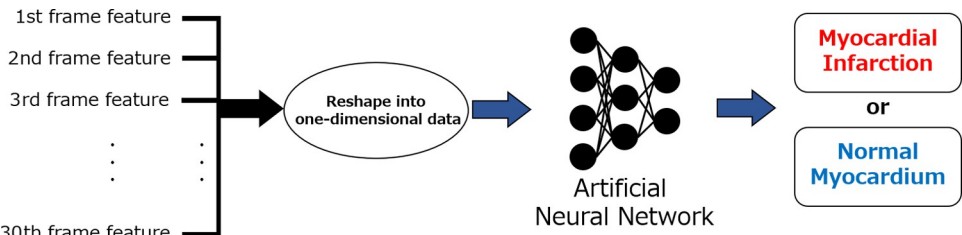

**Fig 11. Schematic representation of the artificial neural network classification of myocardial infarction and normal cases.** Features from 1 to 30 frames were transformed into one-dimensional data and input into an ANN.

and used as input to the neural network to classify the normal and AMI cases. For the hyper-parameters, we set the learning rate to $1 \times 10^{-5}$, the number of epochs to 50, the batch size to 30, and the input data size to $4096 \times 30$.

## Results

Tables 2 and 3 show the confusion matrices and overall classification accuracies of LSTM and ANN for the LX images, while Tables 4 and 5 show the results for the PM images. Table 6 shows the classification accuracy for the LX and PM images for the given parameters and classifiers. Tables 7 and 8 show the sensitivities, specificities, and area under the curves (AUC) for

**Table 2. Overall classification accuracy of long short-term memory for long-axis view images.**

| | | Estimated | | Overall accuracy |
|---|---|---|---|---|
| | | Normal myocardium | Myocardial infarction | |
| Actual | Normal myocardium | 89 | 14 | 0.851 |
| | Myocardial infarction | 16 | 83 | |

**Table 3. Overall classification accuracy of the artificial neural network for the long-axis view images.**

| | | Estimated | | Overall accuracy |
|---|---|---|---|---|
| | | Normal myocardium | Myocardial infarction | |
| Actual | Normal myocardium | 78 | 25 | 0.772 |
| | Myocardial infarction | 21 | 78 | |

Table 4. Overall classification accuracy of the long short-term memory for the short-axis papillary muscle level images.

|  |  | Estimated | | Overall accuracy |
|---|---|---|---|---|
|  |  | Normal myocardium | Myocardial infarction |  |
| Actual | Normal myocardium | 89 | 14 | 0.832 |
|  | Myocardial infarction | 20 | 79 |  |

Table 5. Overall classification accuracy of the artificial neural network for the short-axis papillary muscle level images.

|  |  | Estimated | | Overall accuracy |
|---|---|---|---|---|
|  |  | Normal myocardium | Myocardial infarction |  |
| Actual | Normal myocardium | 85 | 18 | 0.802 |
|  | Myocardial infarction | 22 | 77 |  |

Table 6. Overall classification accuracy with changing parameters.

| View | Classifier | Input parameters for the classifier | | |
|---|---|---|---|---|
|  |  | Output of FC2 | GAP | GMP |
| LX | LSTM | 0.851 | 0.896 | 0.896 |
|  | ANN | 0.772 | 0.817 | 0.817 |
| PM | LSTM | 0.832 | 0.867 | 0.817 |
|  | ANN | 0.802 | 0.797 | 0.837 |

ANN, artificial neural network; FC2, second fully connected layer of VGG16; GAP, global average pooling; GMP, global max pooling; LSTM, long short-term memory; LX, long-axis view; PM, short-axis papillary muscle level

Table 7. Results of long-axis view images.

| View | Classifier | Input parameters for classifier | Sensitivity | Specificity | AUC |
|---|---|---|---|---|---|
| LX | LSTM | Output of FC2 | 0.838 | 0.864 | 0.870 |
|  |  | GAP | 0.879 | 0.913 | 0.943 |
|  |  | GMP | 0.889 | 0.901 | 0.871 |
|  | ANN | Output of FC2 | 0.788 | 0.757 | 0.737 |
|  |  | GAP | 0.788 | 0.845 | 0.871 |
|  |  | GMP | 0.768 | 0.864 | 0.830 |

Table 8. Results of short-axis papillary muscle view images.

| View | Classifier | Input parameters for classifier | Sensitivity | Specificity | AUC |
|---|---|---|---|---|---|
| PM | LSTM | Output of FC2 | 0.798 | 0.864 | 0.883 |
|  |  | GAP | 0.879 | 0.854 | 0.943 |
|  |  | GMP | 0.808 | 0.825 | 0.891 |
|  | ANN | Output of FC2 | 0.777 | 0.825 | 0.796 |
|  |  | GAP | 0.717 | 0.874 | 0.901 |
|  |  | GMP | 0.818 | 0.854 | 0.869 |

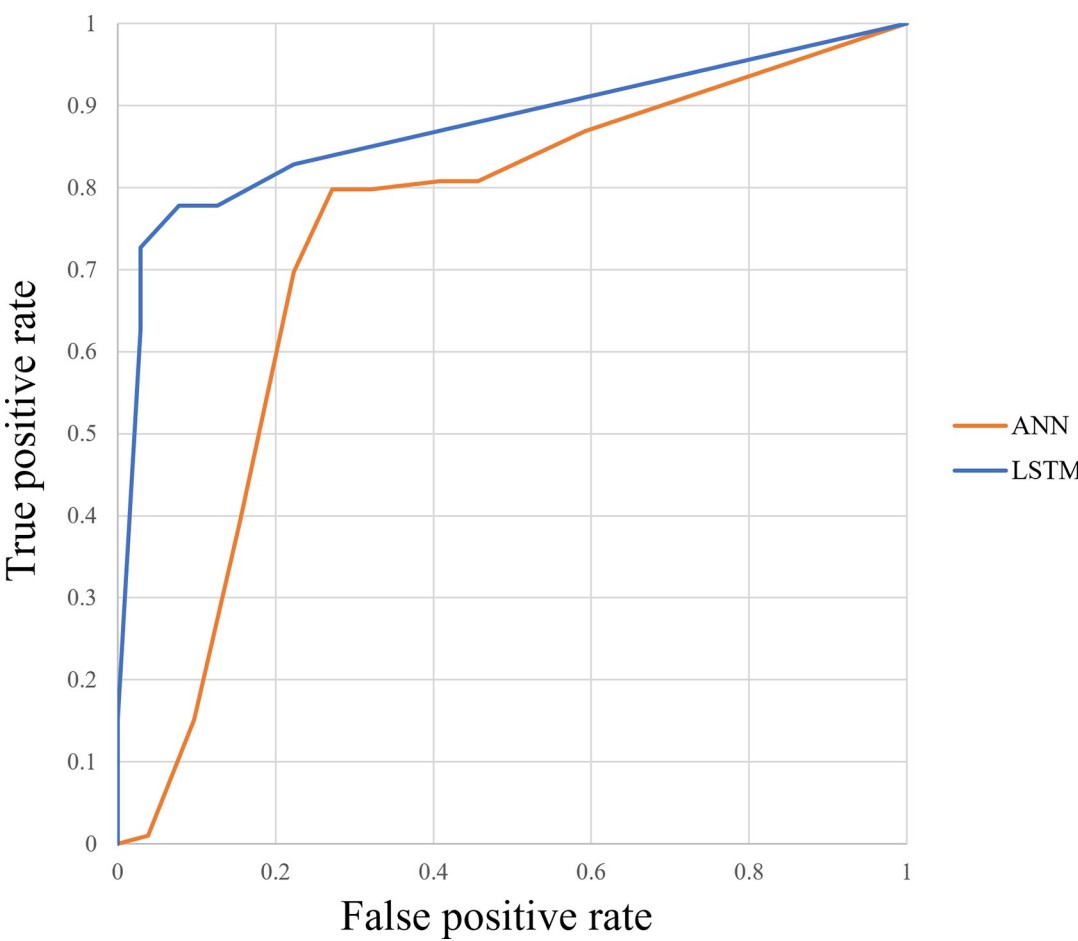

**Fig 12. ROC curves of the LSTM versus ANN in long-axis view.**

the LX and PM images for the given parameters and classifiers. The accuracy of LSTM was the best when GAP was used for both the LX and PM images. The accuracies for the LX and PM images were 0.896 and 0.867, respectively.

Figs 12–15 compare the ROC curve. Figs 16–23 show representative correctly and incorrectly classified cases on echocardiography.

## Discussion

In this study, we proposed an automated classification scheme for AMI and normal cases on echocardiography images using deep learning. The VGG16 method was used to extract features from the echocardiography images, while LSTM was used for the classification. The comparison of the classification models (Tables 2 and 3) shows that the results obtained with LSTM were better than those obtained using the ANN. The overall classification accuracy using LSTM was 0.852 for the LX images and 0.832 for the PM images. These results suggest that LSTM can classify AMI and normal cases with higher accuracy than ANN and analyze and classify useful features over time. In addition, the classification accuracy of LSTM suggests that the image information of one cardiac cycle (consisting of 30 frames) is useful for analyzing myocardial motion, thereby distinguishing AMI from normal myocardium. Unlike ordinary ANNs, LSTMs have a mechanism known as "gates," optimizing them for time-series

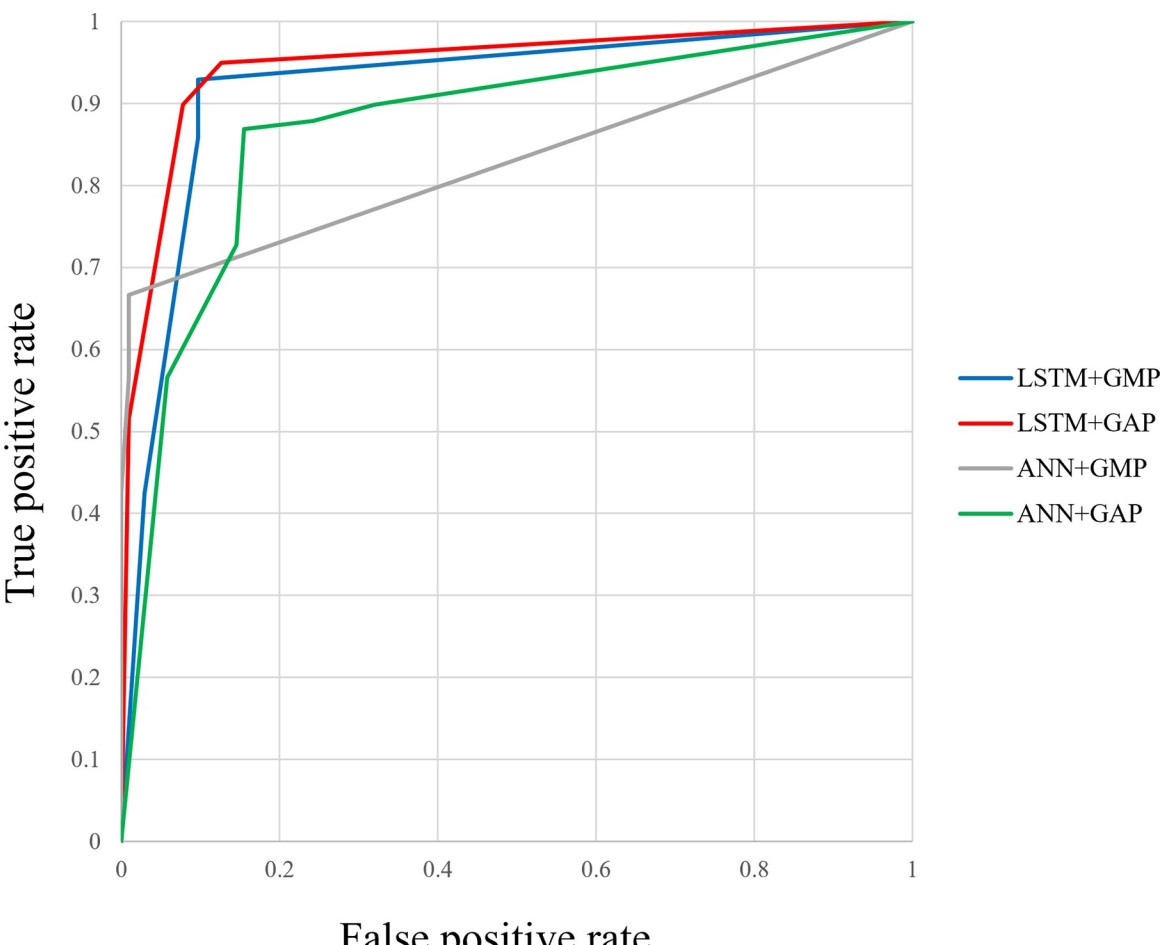

**Fig 13. ROC curves of the GAP versus GMP in long-axis view.**

information analysis. The results showed that the LSTM can detect AMI on echocardiography with better accuracy than ANN by analyzing time series information; moreover, the results confirmed the superiority of the proposed LSTM method. Further, an AI-based study on laryngitis pathology classification showed that the LSTM classification accuracy was 15% higher than regular ANNs [30]. Other studies using AI-based solar radiation prediction have also shown that LSTM has superior accuracy in predicting solar radiation [31]. In this study by comparison, the classification accuracy was improved using LSTM, resulting in more effective use of time series data, once again confirming the effectiveness of this method.

Table 4 shows that the overall classification accuracy of LSTM was best on the LX and PM images when GAP was used. The classification accuracy on the LX images was 0.896, while that on the PM images was 0.867. These results suggest that GAP extracts more useful features for classification than GMP. In addition, the results of the comparison between the features extracted from the fully connected layer and the 512 features extracted by GAP showed that the classification accuracy of LSTM increased when GAP was used. This finding suggests that GAP reduces the number of unnecessary features for classification and achieves more efficient learning by reducing the number of parameters. The reason for the lack of change in classification accuracy between GAP and GMP when LSTM was used in the LX images may be that similar parameters were extracted from the feature maps by GAP and GMP during the pooling process.

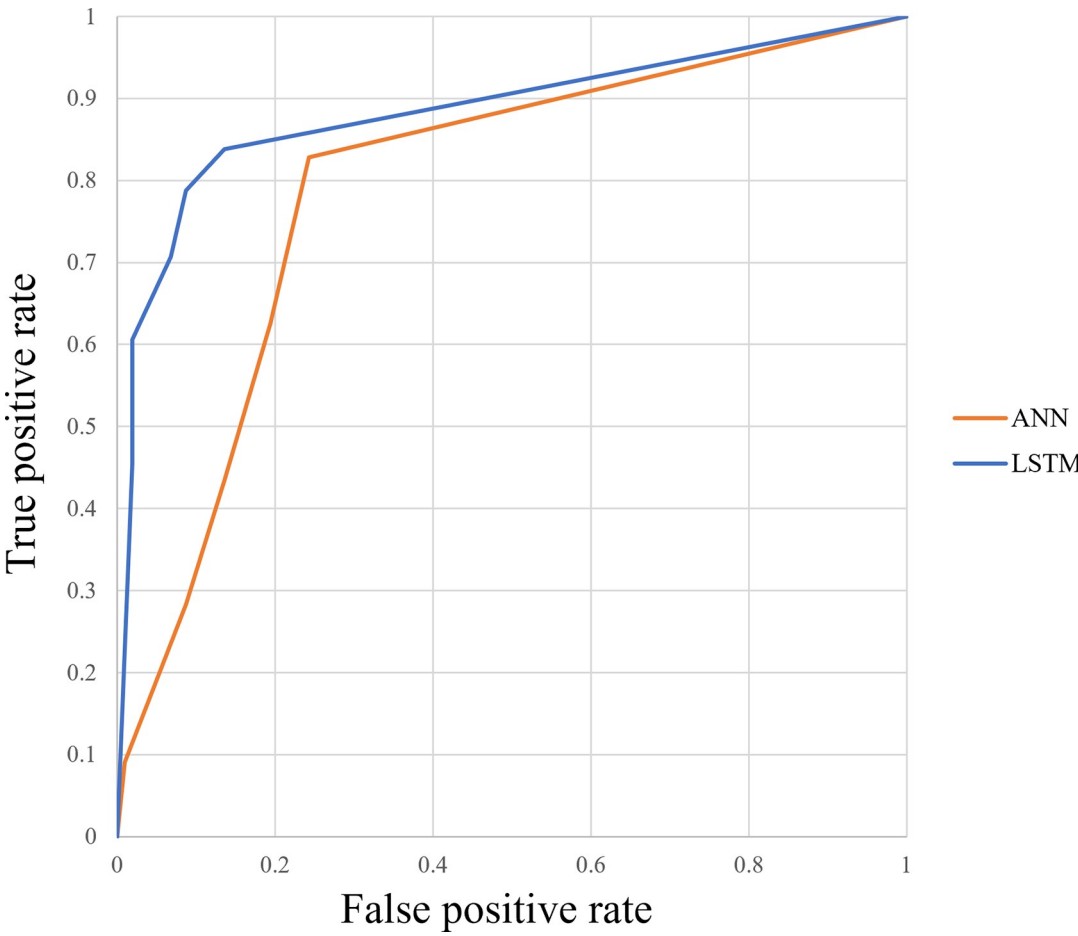

**Fig 14. ROC curves of the LSTM versus ANN in short-axis papillary muscle view.**

Visual comparison of the incorrectly and correctly classified cases showed that those with low video contrast, high noise, or high brightness of the myocardium were misclassified. These results suggest that image quality, such as noise and contrast in the video image, is among the most important factors in the classification of AMI and normal cases on echocardiography. In addition, incorrect cases tended not to be shown adequately in the image: the left ventricle was blurred and a different short-axis level view was shown. In future work, accuracy should be improved by the analysis of echocardiography images and patient data from more facilities to create a robust network model.

Similar studies are listed in Table 9 for comparison with our study. Since there are very few studies with the same images and objectives, a simple comparison with this study may be difficult. However, our method was able to classify MI with an accuracy of more than 80% using 202 cases, confirming its validity.

We then calculated the accuracy of the classification using left ventricular LX and short-axis PM level views, which were subsequently used to detect anteroseptal infarction. Acute anteroseptal infarction was evaluated by cardiologists and experienced sonographers using left ventricular LX and short-axis PM level views as well as apical four-chamber and apical LX views, allowing for observation of the apex. However, inexperienced clinicians, non-cardiologists, residents, and those otherwise unfamiliar with echocardiography may find it difficult to obtain apical four-chamber and apical LX view images with adequate quality. In addition, the

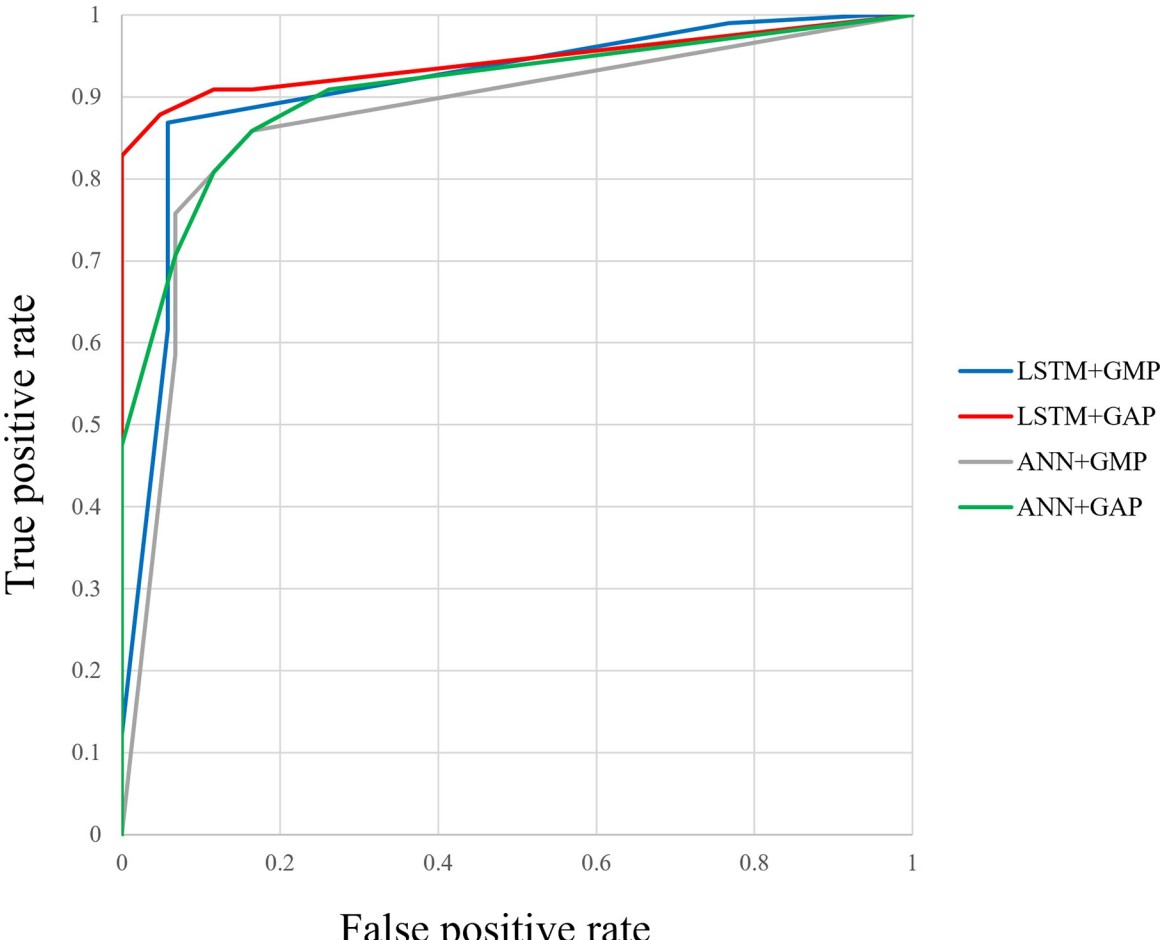

**Fig 15. ROC curves of the GAP versus GMP in short-axis papillary muscle view.**

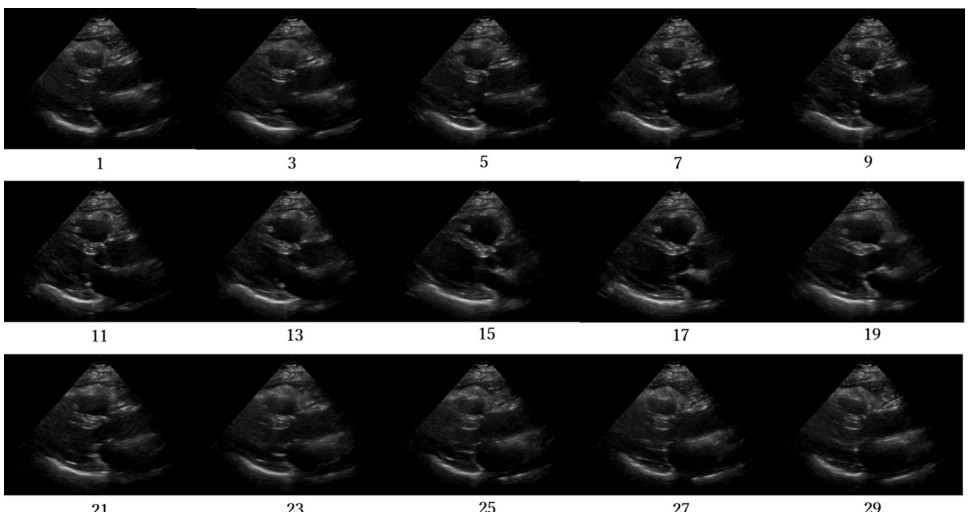

**Fig 16. False-positive cases on long-axis view images.**

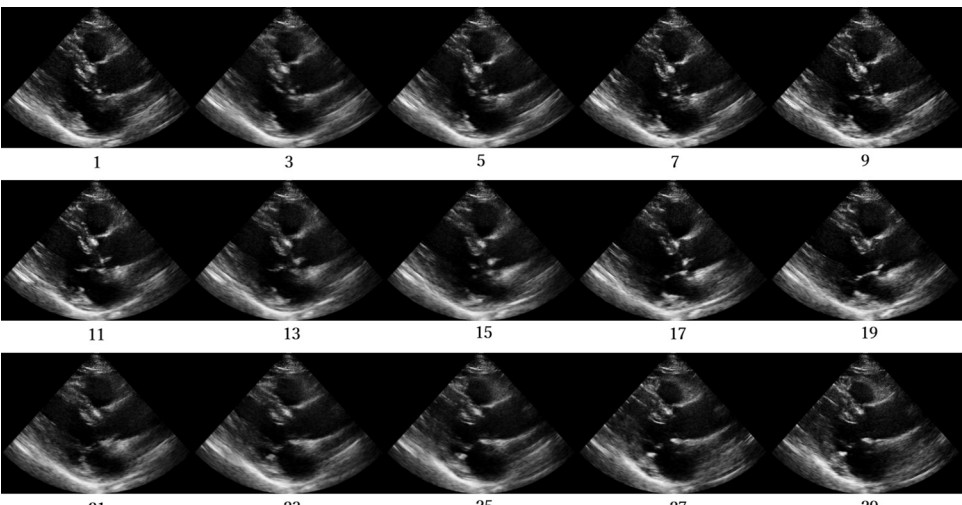

**Fig 17. True-negative cases on long-axis view images.**

detection of AMI is an emergent matter and requires accurate and rapid detection using left ventricular LX and short-axis PM level views, which are relatively easy to obtain. These results indicate that this method correctly identifies acute anteroseptal infarction with superior accuracy clearly distinguishing it from normal myocardium. Therefore, this method can greatly assist non-cardiologists and inexperienced clinicians alike to diagnose acute anteroseptal infarction during initial treatment.

## Limitations and future works

This study has a few limitations. First, the echocardiograms were performed at the same institution. Second, this classification method is performed offline; therefore, it is necessary to apply it to real-time processing so that classification during echocardiography can be utilized. Third, we did not evaluate each segment of the heart individually; rather, we examined only the acute anterior wall septal infarction with occlusion of #6 in the American Heart

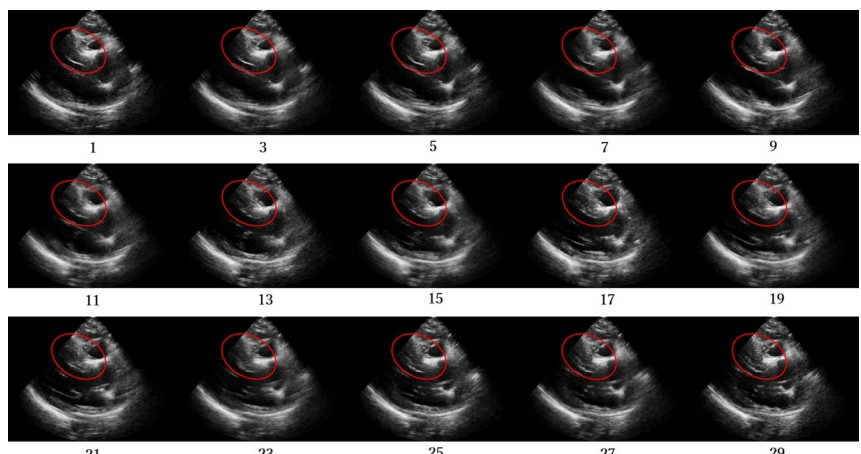

**Fig 18. False-negative cases on long-axis view images with anteroseptal infarction with regional abnormal wall motion circled in red.**

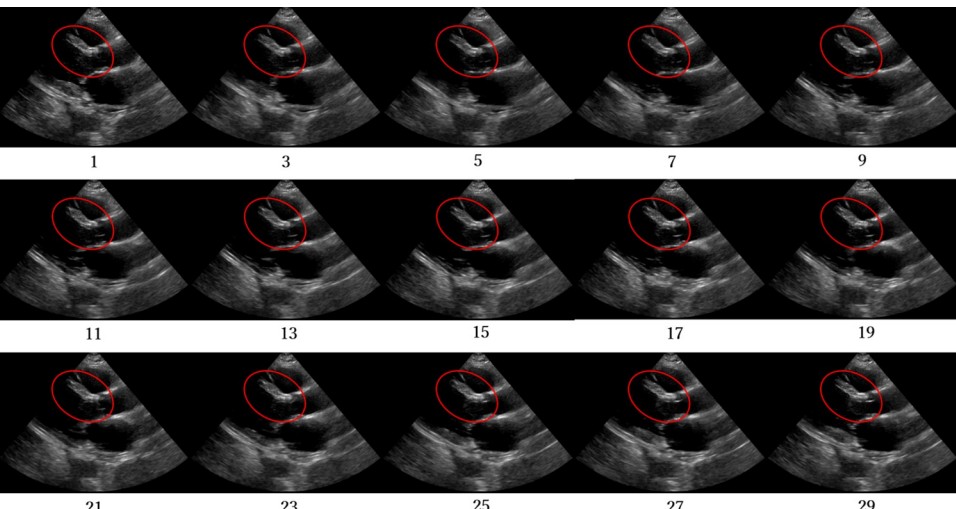

Fig 19. True-positive cases on long-axis view images.

Association Committee Report, which occurs the most frequently. Since this study focused only on acute anteroseptal infarction, the classification and evaluation of infarcts in each segment and in other coronary dominant regions should be performed in the future.

## Conclusion

In this study, we developed an automatic detection scheme for AMI on echocardiography images using CNN and LSTM. The accuracy of the classification showed that our proposed method was able to classify AMI and normal cases with high accuracy, confirming its effectiveness as a supplemental tool for the detection of AMI on echocardiography. Here specialists and skilled doctors can easily detect an anteroseptal infarction. However, it may be difficult for residents and physicians who are unfamiliar with echocardiography at the time of the initial visit or physicians in non-cardiology clinics to detect it. Anteroseptal infarctions occur

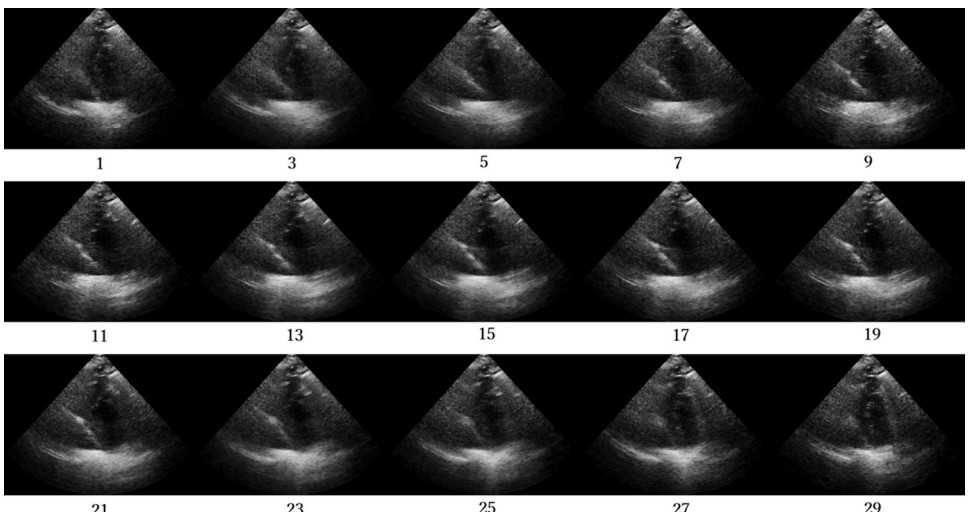

Fig 20. False-positive cases on short-axis view papillary muscle level images.

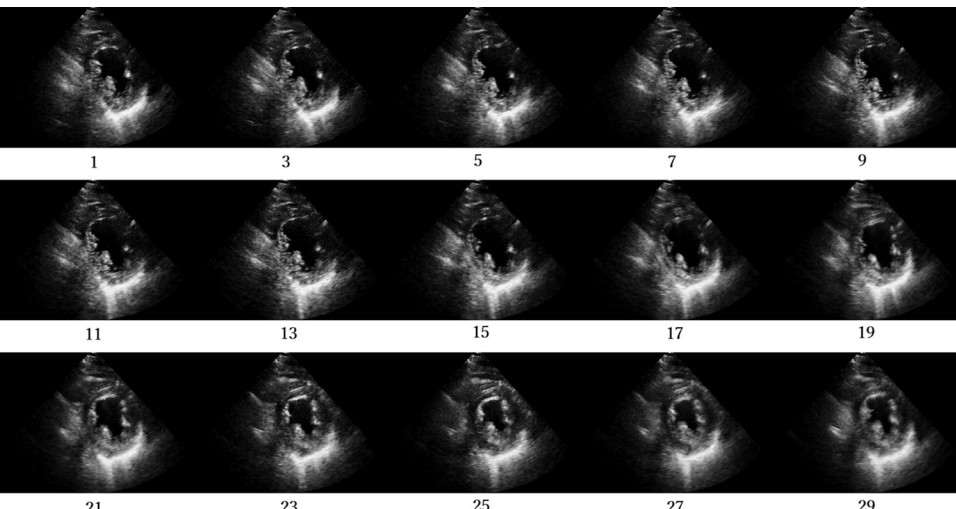

**Fig 21. True-negative cases on short-axis view papillary muscle level images.**

frequently and require accurate detection and diagnosis, regardless of the technician or physician's experience, field, or situation. This method can contribute to the detection of AMI and is expected to lead to its appropriate treatment of and the prognosis of affected patients. Another technical novelty of this study is the use of LSTM, which enables a time-series analysis of wall motion on echocardiography. The results showed that LSTM can detect AMI more accurately than ANNs without a time-series analysis function, confirming its superiority using LSTM. In addition, here we used left ventricular long-axis and short-axis views (papillary muscle level), which are minimal and easy to depict for diagnosis, as input. Since the classifications were performed using highly accurate views, we found the possibility of applying this method using LSTM to other views, which are easy to take. In addition, although this study focused only on acute anteroseptal infarction, its methodology is expected to be extended to the detection of infarcts in other coronary artery dominant regions.

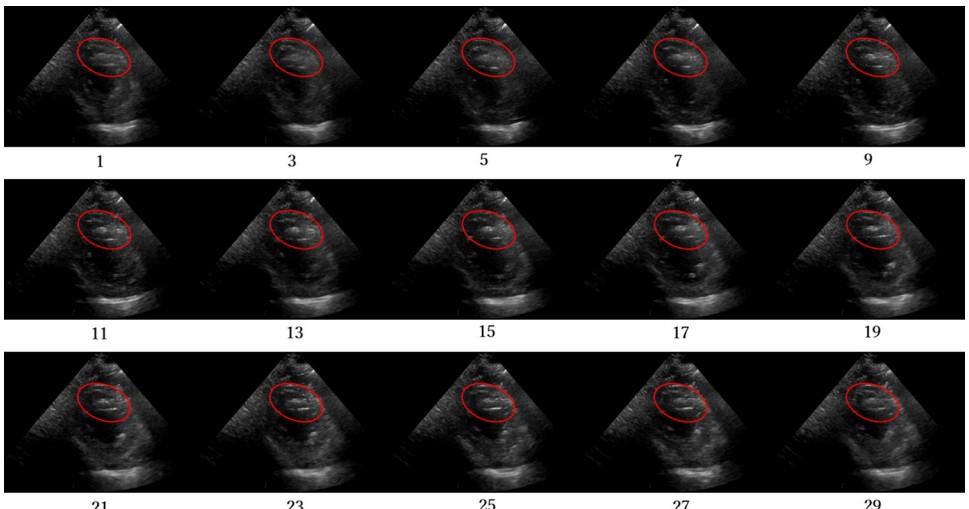

**Fig 22. False-negative cases on short-axis view papillary muscle level images.**

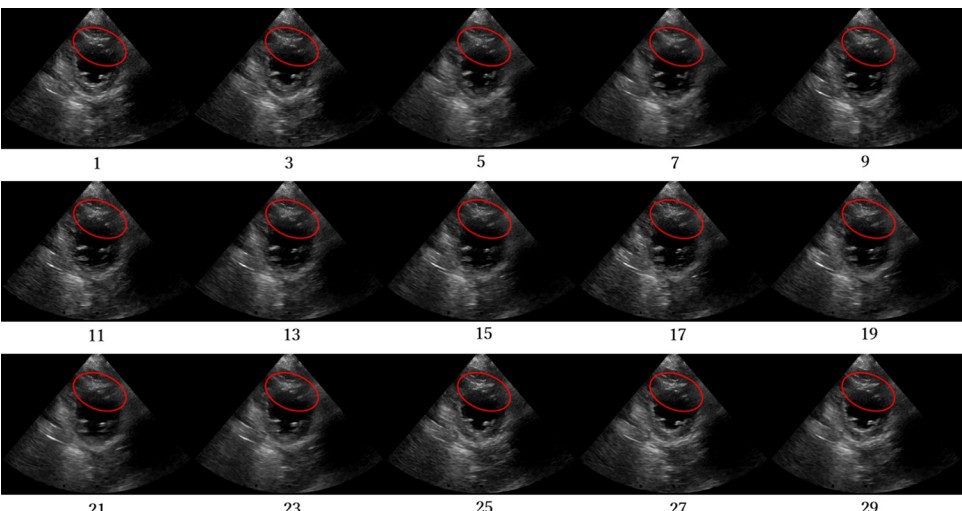

**Fig 23. True-positive cases on short-axis view papillary muscle level images.**

**Table 9. Comparison of related works.**

| Author | dataset | purpose | method | performance |
|---|---|---|---|---|
| Kusunose et al. [7] | Patients with old MIMI: 400 Normal: 100 | Detection of regional abnormal wall motion on echocardiography | Classification using CNN | AUC: 0.97 |
| Zhang et al. [32] | Patients with MIMI: 169 Control: 69 | Detection of chronic MIMI on nonenhanced cardiac cine MRI | Classification using region extraction model and LSTM | Sensitivity: 0.90 Specificity: 0.99 AUC: 0.94 |
| Baloglu et al. [33] | MIMI: 148 Normal: 52 | Detection of MIMI on 12-lead ECG | Classification using CNN | Overall accuracy: 0.9978 |
| Huang et al. [9] | Regional wall motion abnormality: 947 | Detection of regional abnormal wall motion on echocardiography | Segmentation using U-Net | Dice index: 0.756 |
| Vece et al. [34] | Patients with takotsubo syndrome: 110 Patient with a MIAMI: 110 | Detection of Takotsubo cardiomyopathy on echocardiography | Classification using machine learning | AUC: 0.801 Overall accuracy: 0.745 |
| Shimizu et al. [35] | Patients with takotsubo syndrome: 50 Patient with MIAMI: 50 | Detection of Takotsubo cardiomyopathy and AMI on echocardiography | Classification using CNN | Mean accuracy: 0.748 |
| Shahin et al. [36] | 8 of cardioviews: 432 | Classification of echocardiographic view types | Classification using CNN and LSTM | Accuracy: 0.963 |

## Acknowledgments

We thank Mr. Kimura and Ms. Yamazaki for the useful discussions as well as Prof. Saito and Prof. Fujita for assisting and teaching.

## Author Contributions

**Conceptualization:** Atsushi Teramoto, Eiichi Watanabe.

**Data curation:** Ryosuke Muraki, Keiko Sugimoto, Kunihiko Sugimoto, Eiichi Watanabe.

**Formal analysis:** Ryosuke Muraki.

**Funding acquisition:** Eiichi Watanabe.

**Investigation:** Ryosuke Muraki, Atsushi Teramoto, Keiko Sugimoto, Kunihiko Sugimoto, Akira Yamada, Eiichi Watanabe.

**Methodology:** Ryosuke Muraki, Atsushi Teramoto.

**Project administration:** Atsushi Teramoto, Eiichi Watanabe.

**Software:** Ryosuke Muraki.

**Validation:** Akira Yamada.

**Writing – original draft:** Ryosuke Muraki.

**Writing – review & editing:** Atsushi Teramoto, Keiko Sugimoto, Kunihiko Sugimoto, Akira Yamada, Eiichi Watanabe.

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
