## [Decision Letter · Decision Letter 0]

31 May 2021

PONE-D-21-16074

Automated detection scheme for acute myocardial infarction using convolutional neural network and long short-term memory

PLOS ONE

Dear Dr. Teramoto,

Thank you for submitting your manuscript to PLOS ONE. After careful consideration, we feel that it has merit but does not fully meet PLOS ONE’s publication criteria as it currently stands. Therefore, we invite you to submit a revised version of the manuscript that addresses the points raised during the review process.

 Based on the comments received by the reviewer and my own observations, I recommend major revisions for the paper. The authors have to address all the suggestions/comments from the reviewers carefully.

We look forward to receiving your revised manuscript.

Kind regards,

Thippa Reddy Gadekallu

Academic Editor

PLOS ONE

Journal Requirements:

2. In ethics statement in the manuscript and in the online submission form, please provide additional information about the patient records used in your retrospective study. Specifically, please ensure that you have discussed whether all data were fully anonymized before you accessed them and/or whether the IRB or ethics committee waived the requirement for informed consent. If patients provided informed written consent to have data from their medical records used in research, please include this information.

3. Please include the data sources used in the Data availability statement and Methods section. We note that the origin of the testing dataset does not seem to be referenced.

4.  Thank you for including your ethics statement:  "This study was approved by an institutional review board and informed consents were obtained from patients subject to the condition of data anonymization (No. HM20).".   

Reviewers' comments:

Reviewer's Responses to Questions

**Comments to the Author**

1. Is the manuscript technically sound, and do the data support the conclusions?

Reviewer #1: Yes

Reviewer #2: Yes

2. Has the statistical analysis been performed appropriately and rigorously? 

Reviewer #1: Yes

Reviewer #2: Yes

3. Have the authors made all data underlying the findings in their manuscript fully available?

Reviewer #1: Yes

Reviewer #2: Yes

4. Is the manuscript presented in an intelligible fashion and written in standard English?

Reviewer #1: Yes

Reviewer #2: Yes

5. Review Comments to the Author

Reviewer #1: - Spell out each acronym the first time used in the Abstract as well as the body of the paper.

- Make sure all figures and tables are referred to in the text.

- There are many typos and grammatical mistakes in the entire paper.

- There must be column names in the tables with all borders.

- All the key terms of the equations must be mentioned

- Qualities of figures are not good.

- Relevant literature review of latest similar research studies on the topic at hand must be discussed

- The summary at the end of the literature review should be focused on the limitations of related work.

- The authors should clearly mention all parameters used to evaluate the performance of the classifiers.

- Authors can add below references:

1) Rehman, A., Rehman, S. U., Khan, M., Alazab, M., & Reddy, T. (2021). CANintelliIDS: Detecting In-Vehicle Intrusion Attacks on a Controller Area Network using CNN and Attention-based GRU. *IEEE Transactions on Network Science and Engineering*.

2) Abbas, S., Jalil, Z., Javed, A. R., Batool, I., Khan, M. Z., Noorwali, A., ... & Akbar, A. (2021). BCD-WERT: a novel approach for breast cancer detection using whale optimization based efficient features and extremely randomized tree algorithm. *PeerJ Computer Science*, *7*, e390.

Reviewer #2: 1. What are the limitations of existing works that motivated the current research?

2. What is the novelty of the work?

3 List out the main contributions of the current work?

4. SUmmarize the related works in the form of a table.

5. Some of the recent works on LSTM and CNN applied in different domains such as the following can be discussed in the paper: "Hand gesture classification using a novel CNN-crow search algorithm, A multidirectional LSTM model for predicting the stability of a smart grid, Image-Based malware classification using ensemble of CNN architectures (IMCEC)".

6. COmpare the results obtained with recent state-of-the-art.

7. Discuss about the limitations and future scope of the current work.

6. PLOS authors have the option to publish the peer review history of their article (what does this mean?). If published, this will include your full peer review and any attached files.

Reviewer #1: No

Reviewer #2: No

---

## [Author Response · Author response to Decision Letter 0]

29 Aug 2021

Dear Editor and Reviewers:

We thank you and the reviewers for your thoughtful suggestions and insights. The manuscript has benefited from these insightful suggestions. I look forward to working with you and the reviewers to move this manuscript closer to publication in PLOS ONE.

The manuscript has been rechecked and the necessary changes have been made in accordance with the reviewers’ suggestions. The responses to all the comments have been provided and the changes in the manuscript are indicated in red font. 

In detail, please refer to the response document and revised manuscript.

Thank you for your consideration. I look forward to hearing from you.

Sincerely,

Atsushi Teramoto, Ph.D.

---

## [Decision Letter · Decision Letter 1]

13 Oct 2021

PONE-D-21-16074R1Automated detection scheme for acute myocardial infarction using convolutional neural network and long short-term memoryPLOS ONE

Dear Dr. Teramoto,

Thank you for submitting your manuscript to PLOS ONE. After careful consideration, we feel that it has merit but does not fully meet PLOS ONE’s publication criteria as it currently stands. Therefore, we invite you to submit a revised version of the manuscript that addresses the points raised during the review process.

The authors are requested to revise their manuscript and address all the comments raised by Reviewer 4.

We look forward to receiving your revised manuscript.

Kind regards,

Binh P. Nguyen, Ph.D.

Academic Editor

PLOS ONE

Journal Requirements:

Additional Editor Comments (if provided):

Reviewers' comments:

Reviewer's Responses to Questions

**Comments to the Author**

1. If the authors have adequately addressed your comments raised in a previous round of review and you feel that this manuscript is now acceptable for publication, you may indicate that here to bypass the “Comments to the Author” section, enter your conflict of interest statement in the “Confidential to Editor” section, and submit your "Accept" recommendation.

Reviewer #3: All comments have been addressed

Reviewer #4: (No Response)

2. Is the manuscript technically sound, and do the data support the conclusions?

Reviewer #3: Yes

Reviewer #4: Yes

3. Has the statistical analysis been performed appropriately and rigorously? 

Reviewer #3: Yes

Reviewer #4: Yes

4. Have the authors made all data underlying the findings in their manuscript fully available?

Reviewer #3: No

Reviewer #4: Yes

5. Is the manuscript presented in an intelligible fashion and written in standard English?

Reviewer #3: Yes

Reviewer #4: Yes

6. Review Comments to the Author

Reviewer #3: I believe there were a few good questions have been asked and authors have responded and addressed all previous comments.

Reviewer #4: Thank you for the opportunity to review ‘Automated detection scheme for acute myocardial infarction using convolutional neural network and long short-term memory” by Muraki et al. In this paper, the authors investigated the feasibility of automatic detection method for acute myocardial infarction using convolutional neural networks (CNNs) and long short-term memory (LSTM) in echocardiography. A total of 202 cines were obtained with 99 diagnoses of acute anteroseptal infarction 106 normal cases. The authors supported the usefulness of the CNNs and LSTM method for the detection of myocardial infarction using echocardiography. The following are my comments on the manuscript:

1. The main limitation of this study is only one segment (anteroseptal infarction) was included, which may not be clinically applicable. Further, the authors included the myocardial infarction with proximal left anterior descending (LAD) artery occlusion which normally results in multiple abnormal segments. I am wondering that the authors take other segments into the account.

2. Could the authors provide the criteria for myocardial infarction with proximal LAD occlusion in this study? Was coronary angiography performed for all patients to confirm the proximal LAD lesions?

3. It would be much better if the authors can provide the time of echocardiography captured: is it before or after revascularization? Noticeably, echocardiograms were obtained after revascularization, the regional wall motion abnormalities may return to normal.

3. Most importantly, the occlusion of proximal LAD will result in multiple abnormal segments which are easily detected by echocardiogram in clinical practice. I am wondering how significantly this study contributes to clinical practice.

5. As a clinical physician, I would like to know the sensitivity, specificity, and area under the curve of this technique. These metrics will provide physicians deeper insights into this technique.

6. Lastly, it would suggest that the authors provide Table 1: baseline clinical characteristics of this cohort which will illustrate the target population that this technique is applying on.

7. PLOS authors have the option to publish the peer review history of their article (what does this mean?). If published, this will include your full peer review and any attached files.

Reviewer #3: No

Reviewer #4: No

---

## [Author Response · Author response to Decision Letter 1]

22 Nov 2021

Please refer the response document.

---

## [Decision Letter · Decision Letter 2]

2 Feb 2022

Automated detection scheme for acute myocardial infarction using convolutional neural network and long short-term memory

PONE-D-21-16074R2

Dear Dr. Teramoto,

We’re pleased to inform you that your manuscript has been judged scientifically suitable for publication and will be formally accepted for publication once it meets all outstanding technical requirements.

Kind regards,

Binh P. Nguyen, Ph.D.

Academic Editor

PLOS ONE

Additional Editor Comments (optional):

Reviewers' comments:

Reviewer's Responses to Questions

**Comments to the Author**

1. If the authors have adequately addressed your comments raised in a previous round of review and you feel that this manuscript is now acceptable for publication, you may indicate that here to bypass the “Comments to the Author” section, enter your conflict of interest statement in the “Confidential to Editor” section, and submit your "Accept" recommendation.

Reviewer #4: All comments have been addressed

2. Is the manuscript technically sound, and do the data support the conclusions?

Reviewer #4: Yes

3. Has the statistical analysis been performed appropriately and rigorously? 

Reviewer #4: Yes

4. Have the authors made all data underlying the findings in their manuscript fully available?

Reviewer #4: Yes

5. Is the manuscript presented in an intelligible fashion and written in standard English?

Reviewer #4: Yes

6. Review Comments to the Author

Reviewer #4: (No Response)

7. PLOS authors have the option to publish the peer review history of their article (what does this mean?). If published, this will include your full peer review and any attached files.

Reviewer #4: No

---

## [Editor Report · Acceptance letter]

10 Feb 2022

PONE-D-21-16074R2 

Automated detection scheme for acute myocardial infarction using convolutional neural network and long short-term memory 

Dear Dr. Teramoto:

I'm pleased to inform you that your manuscript has been deemed suitable for publication in PLOS ONE. Congratulations! Your manuscript is now with our production department. 

Kind regards, 

on behalf of

Dr. Binh P. Nguyen 

Academic Editor

PLOS ONE